# Neutrophil Extracellular Traps in Airway Diseases: Pathological Roles and Therapeutic Implications

**DOI:** 10.3390/ijms24055034

**Published:** 2023-03-06

**Authors:** Ara Jo, Dae Woo Kim

**Affiliations:** Department of Otorhinolaryngology-Head & Neck Surgery, Boramae Medical Center, Seoul National University College of Medicine, Seoul 07061, Republic of Korea

**Keywords:** neutrophil extracellular traps, therapeutic targets, airway diseases, asthma, chronic rhinosinusitis, cystic fibrosis, chronic obstructive pulmonary disease, bronchiectasis, coronavirus disease 2019

## Abstract

Neutrophils are important effector cells of the innate immune response that fight pathogens by phagocytosis and degranulation. Neutrophil extracellular traps (NETs) are released into the extracellular space to defend against invading pathogens. Although NETs play a defensive role against pathogens, excessive NETs can contribute to the pathogenesis of airway diseases. NETs are known to be directly cytotoxic to the lung epithelium and endothelium, highly involved in acute lung injury, and implicated in disease severity and exacerbation. This review describes the role of NET formation in airway diseases, including chronic rhinosinusitis, and suggests that targeting NETs could be a therapeutic strategy for airway diseases.

## 1. Introduction

Neutrophils, which are the most abundant innate immune cells, have a short lifespan and serve as the first defensive response against invading pathogens [1]. Neutrophils are produced in the bone marrow, and their release is regulated by C–X–C motif chemokine ligand 8 (CXCL8)-mediated neutrophil mobilization, which allows neutrophils to arrive at sites of inflammation [2]. They are characterized by a multilobed nucleus and granular cytoplasm, which are important for host defense. In the antibacterial response, neutrophils kill pathogens by degranulation, phagocytosis, cytokine production, and neutrophil extracellular trap (NET) formation [3,4]. Numerous studies have revealed that neutrophils play a central role in chronic inflammatory conditions, such as cancer [5,6], autoimmune diseases [7,8,9], and airway diseases [10,11]. NETs are net-like structures that kill pathogens by forming an extracellular structure with chromatin and granule proteins, such as neutrophil elastase (NE), myeloperoxidase (MPO), and calprotectin [12,13]. NET formation, also called NETosis, is a form of neutrophil cell death distinct from apoptosis and necroptosis, and its antimicrobial activity has been described [13,14]. NETs can trap almost any type of pathogens, even those that are too large to phagocytose, and they play an important role in the host’s defense against infections that evade normal neutrophil killing [15,16]. Although NETs serve antibacterial functions, they are also known to be involved in the pathogenesis of multiple diseases and in disease severity [17,18,19,20]. This review describes the roles of NETs in airway diseases and suggests that a better understanding of their role may be crucial for therapeutic treatments. The systematic review was conducted in accordance with Preferred Reporting Items for Systematic Reviews and Meta-Analyses (PRISMA) guidelines (Appendix A).

## 2. Neutrophil Extracellular Traps (NETs)

During NET formation, nuclear and granular membranes dissolve, and the chromatin decondenses in the cytoplasm. Subsequently, the plasma membrane is ruptured and releases extracellular structures with DNA and several granule proteins, which serve to trap and kill pathogens (Figure 1). NETs are considered a double-edged sword of the immune system; they can play a beneficial role in innate immunity, but they can also exert proinflammatory effects and cause tissue damage [21,22].

NETs are composed of extracellular DNA (eDNA) fibers with histones and granule proteins, including NE, MPO, calprotectin, and α-defensins. These components are released into the extracellular space after the cell membrane ruptures and induce the secretion of proinflammatory cytokines, which are implicated in airway neutrophilia and the airway inflammasome [23,24].

Histones are the major protein components of the nucleosome, which consists of an octamer of core histones (H2A, H2B, H3, and H4) wrapped in superhelical DNA. Histones can induce cytotoxicity through cell swelling and release of lactate dehydrogenase, cytokines, and chemokines, and may be involved in lung damage [25,26,27]. Some studies have reported that extracellular histones activate neutrophils to induce NET release [28,29]. NE is a serine protease stored in the azurophilic granules of neutrophils and acts as a host defense mechanism in inflammation, the immune response, and coagulation [30]. Nicotinamide adenine dinucleotide phosphate (NADPH) oxidase generates reactive oxygen species (ROS) production. The generation of superoxide is mediated by NADPH oxidase complex and converted to hydrogen peroxide. MPO uses hydrogen peroxide as a substrate and promotes the release of NE from neutrophil granules. MPO and NE translocate from neutrophil granules into the nucleus and trigger the release of NETs [31]. MPO binds chromatin with NE and contributes to the decondensation of chromatin, and then the DNA is released into the extracellular space to capture invading pathogens [32,33]. NE produces several inflammatory chemokines, such as interleukin (IL)-8 and matrix metalloproteinase (MMP)-9 [34,35]. The inhibition of NE activity reduces tissue damage by attenuating inflammation and preventing NET formation [36]. Imbalances between NE and its inhibitors are implicated in the pathogenesis of several diseases [37,38]. MPO levels were elevated in patients with asthma and correlated with NET-derived proteins [23,39]. MPO/DNA complexes were also present at higher levels in patients with chronic inflammatory diseases and were associated with disease severity and clinical outcomes [40,41,42]. Calprotectin is a noncovalent heterodimer complex of calcium and zinc-binding protein of the S-100 family (S100A8 and S100A9) that is found primarily in neutrophils. Elevated calprotectin levels were found in inflammatory diseases and correlated strongly with disease severity [43,44]. α-Defensins, which are neutrophil granule proteins, are known to be neutrophil-associated antimicrobial peptides important for mucosal immune protection. Elevated levels of α-defensins and LL-37 were found in neutrophilic phenotype of airway diseases and were associated with cytotoxicity [45,46].

LL-37, the only type of cathelicidin present in humans [47], which is also an antimicrobial protein, contributes to disruption of the nuclear membrane and induces NET formation [48]. LL-37 was positively correlated with IL-8, tumor necrosis factor (TNF)-α, and the percentage of neutrophils [49]. Increased levels of LL-37 have been reported in severe disease and exacerbations [49,50]. MMPs are zinc-dependent endoproteases that are responsible for the degradation of extracellular matrix proteins, including collagen and fibronectin. MMP-8 (neutrophil collagenase) and MMP-9 (gelatinase B) are usually found in neutrophils, and their release induces NET formation [51]. MMP levels are elevated in neutrophilic airway diseases and are inversely correlated with disease severity and lung function [52,53,54]. Protein arginine deiminase type 4 (PAD4) is activated by protein kinase C (PKC) and translocates into the nucleus to induce histone citrullination [55,56]. It contributes to chromatin decondensation, leading to NET formation [57]. Excessive PAD4 activity has been related to airway disease. PAD4 inhibition or deficiency reduces citrullinated histones and NET-induced lung injury [58,59,60].

NETs are triggered by various stimuli, such as pathogens, proinflammatory cytokines, antibodies, and chemical stimuli, such as phorbol myristate acetate (PMA) [61,62,63]. Lipopolysaccharide (LPS) is an important outer membrane component of Gram-negative bacteria. LPS triggers inflammatory responses through Toll-like receptor 4 (TLR4), activates neutrophils, and promotes NET formation [13,64]. PMA is used to trigger NET formation. PMA activates PKC, which in turn activates NADPH oxidase and induces ROS production and calcium flux within the cell, leading to NET formation [65,66]. IL-8, also known as CXCL8, is a proinflammatory chemokine that attracts and activates neutrophils and triggers NET formation [13]. IL-8 is expressed in a variety of cell types, including neutrophils. Elevated IL-8 levels have been found in various airway diseases [67,68,69] and cause tissue damage and lung dysfunction [70].

Autophagy is primarily considered as a cell survival process to limit cellular damage. However, increased neutrophil autophagy and NET formation were associated with asthma severity and airway epithelial damage [21]. Qu et al. demonstrated that NET-treated alveolar epithelial cells induced in abnormal autophagy resulted in cellular damage [71]. A recent study reported that NETs translocate from the liver to the lungs by blood and promote acute lung injury [72]. NETs have been widely reported to induce chronic inflammation and to be involved in disease pathogenesis. NETs appear to be a potential biomarker for patients with airway diseases, but more research is needed to understand this phenomenon.

## 3. NETs in Airway Diseases

Investigators have described mechanisms of NET formation, as well as the role of NETs in several diseases, such as chronic inflammatory diseases, autoimmune diseases, and metabolic disorders. Although NETs play a role in host defense, NETs induce direct cytotoxic effects on the lung epithelium and endothelium [73]. NET formation also causes many indirect complications, such as airway obstruction of the respiratory system. Excessive NETs are associated with numerous chronic inflammatory diseases, such as asthma [42,74,75], chronic obstructive pulmonary disease (COPD) [74,75,76], and cystic fibrosis (CF) [74,75,77] (Figure 2).

### 3.1. Asthma

Asthma is a heterogeneous chronic inflammatory disease of the airways characterized by airway hyper-responsiveness, airflow obstruction, and airway remodeling. There are several subtypes of asthma, with different endotypes and phenotypes [78]. Asthma patients are divided into four inflammatory phenotypes according to the cellular composition of the sputum: eosinophilic, neutrophilic, paucigranulocytic, or mixed granulocytic. Adult asthma is primarily associated with type 2 inflammation, characterized by the expression of type 2 cytokines (IL-4, IL-5, and IL-13), eosinophilic airway infiltration, and mucus secretion [79]. On the other hand, type 2-low asthma is characterized by neutrophilic or paucigranulocytic inflammation and is associated with severity and corticosteroid resistance [80,81,82]. NETs are known to be beneficial for host defense, but they damage airway epithelial cells and induce autoantigen production in airway epithelial cells [21,83]. Several studies have demonstrated that NET levels were elevated in patients with severe asthma [23,42,45]. IL-8 is a potent NET inducer and has been found to be elevated in severe asthma [84,85,86]. Abnormally high levels of neutrophils, NE, and IL-8 were observed in patients with neutrophilic asthma [87]. Pham et al. reported that IL-8 induced neutrophil autophagy and NET production in patients with severe asthma [21]. Neutrophil-derived MMP-9 is implicated in asthma severity [88]. Elevated levels of NET components, such as eDNA, LL-37, α-defensins, and NE, were found in patients with neutrophilic asthma and associated with worsened lung function and asthma severity [45]. eDNA was associated with elevated expression of IL-8 and IL-1β, as well as increased caspase-1 activity [45]. This may suggest that NETs can activate the inflammasome to trigger the secretion of IL-1β [23,89,90]. Increased calprotectin levels were detected in the serum of asthma patients and correlated with the neutrophil proportion and lung function decline [91]. Neutrophil cytoplasts, which are enucleated cell bodies released during NETosis, correlated positively with IL-17 levels and were detected in severe asthma with neutrophil inflammation [24]. In a neutrophilic asthma animal model, NETs caused severe airway inflammation, and inhibition of NET production reduced airway inflammation and hyperresponsiveness [92]. This evidence proves that NETs contribute to the pathophysiology of asthma; therefore, it is necessary to understand the various mechanisms of NET formation.

### 3.2. Chronic Rhinosinusitis (CRS)

CRS is a heterogeneous inflammatory disease characterized by persistent symptomatic inflammation of the nose and paranasal sinuses [93], caused by multifactorial factors [94,95]. CRS is classified into CRS with nasal polyps (CRSwNP) and without nasal polyps (CRSsNP) according to the phenotype, and type 2 and non-type 2 according to the endotype [96]. Type 2 inflammation with eosinophils is predominant in Western countries, whereas non-type 2 inflammation with neutrophils is found in Asian countries [97,98]. Neutrophil infiltration has been found in the sinus mucosa of CRS patients, suggesting that neutrophils may play an important role in the pathologic process of CRS [99]. In addition, neutrophilic inflammation is associated with corticosteroid resistance in CRSwNP [100,101], which may be accounted for by the IL-36γ/IL-36R pathway [101]. Mucus levels of IL-1β, IL-6, IL-8, and TNF-α were significantly higher in elderly (≥60 years) CRS patients and were positively correlated with neutrophilia [102]. It has been suggested that elderly CRS patients tend to have neutrophilic inflammation and may be less responsive to corticosteroids. Kim et al. demonstrated that tissue neutrophilia is a risk factor for refractory CRSwNP in an Asian population and found that elevated expression of NE and IL-36α was highly associated with refractoriness [103]. Furthermore, tissue neutrophil and NET expression were correlated with refractoriness in non-eosinophilic or neutrophilic CRSwNP [104]. NETs were present in the sinus mucosa and subepithelial layer, and were significantly higher in CRSwNP [46]. Neutrophilic granule proteins, such as MPO and NE, as well as antimicrobial proteins such as LL-37, showed significantly increased expression in CRSwNP patients [46,103,105]. MPO was significantly higher in tissues from patients with recurrent CRSwNP [105]. In addition, eDNA and NET-forming neutrophils were elevated in CRS regardless of the presence of nasal polyps; interestingly, their levels were significantly higher in nasal secretions from patients with exacerbated CRSwNP or exacerbated CRSsNP [106]. Calprotectin (S100A8/A9) levels were significantly higher in NP tissue from CRSwNP patients than in controls and were highly correlated with tissue IL-8 and MPO levels both in controls and CRSwNP patients [107]. Interestingly, Delemarre et al. reported that the number of neutrophils, the expression of MPO, IL-6, and IL-8, and the activity of cathepsin G/NE were elevated in patients with type 2 CRSwNP. They also observed the coexistence of eosinophilic and neutrophilic inflammation in severe type 2 CRSwNP [108]. Cluster analysis showed high expression levels of neutrophil-associated MPO, IL-8, IL-6, IL-1β, MMP-8, and MMP-9 in CRS with low type 2 inflammation [109]. NETs were also found to be abundant in the NP tissues of CRS patients with low type 2 inflammation, and eosinophil extracellular traps (EETs) were rarely observed. In cases of high type 2 inflammation with the highest levels of IL-5, eosinophil cationic protein, and total immunoglobulin E, eosinophil extracellular traps were predominant, and few NETs were observed [109]. Several reports have been published on NET formation in CRS, but there are fewer studies than on other diseases; thus, further research is needed.

### 3.3. Cystic Fibrosis (CF)

CF is an inherited chronic inflammatory disease characterized by neutrophilic-dominant inflammation. Free DNA and neutrophil counts were significantly higher in the bronchoalveolar lavage fluid (BALF) of CF patients compared to healthy controls [110], and other studies have suggested that this DNA is derived from NETs [111,112]. Elevated eDNA levels were found in the sputum and BALF from CF patients and were associated with worsened lung function [111,113,114]. Several studies have demonstrated that NET formation was increased in airway samples from CF patients and was associated with worsened lung function [77]. NET components, including NE and MPO, were present in the sputum of CF patients [113,115]. NE is a risk factor for severity and directly induces structural lung damage in CF patients [115,116]. MPO is associated with worse clinical outcomes and lung function decline in CF [117]. MMP-8 and MMP-9 levels were elevated in the BALF of CF patients [118] and may contribute to the progression of CF [119]. Another component of NETs, calprotectin (S100A8/A9), is negatively correlated with severity of obstruction, as measured by parameters such as the forced expiratory volume in 1 s (FEV1) [120,121]. Autoantibodies against bactericidal permeability-increasing protein (BPI), which is stored in azurophilic granules of neutrophils, are found in CF patients, associated with NETs and worsened lung function [122]. This may be further evidence that NETs are involved in autoimmunity in CF. Delayed neutrophil apoptosis led to a longer neutrophil lifespan and increased NET production [123]. Recent studies have suggested that NETs are a major factor in lung inflammation and damage in CF. Therefore, it is necessary to develop NET-targeted therapies that yield positive effects, while minimizing side-effects such as cytotoxicity.

### 3.4. Chronic Obstructive Pulmonary Disease (COPD)

Chronic obstructive pulmonary disease (COPD) is a progressive pulmonary disease that is characterized by progressive inflammation and airflow limitation. Neutrophilic inflammation is a hallmark of COPD after long-term exposure to external stresses, such as cigarette smoke, viruses, and oxidative stress [124,125]. Several studies have highlighted that airway neutrophil levels are associated with disease severity and exacerbations, as well as corticosteroid resistance in COPD [126,127,128]. Neutrophils are a major source of IL-17, and sputum IL-17 levels were associated with airflow limitation and obstruction in COPD [129,130]. NETs have also been found in sputum from both stable-state and exacerbated COPD patients [131,132,133], and their concentration was associated with disease severity in patients with COPD [19]. The expression of sputum IL-8, MMP-9, and NE showed positive correlations with the neutrophil proportion, while lung function showed a negative correlation [128,134,135]. NET components, including eDNA, LL-37, α-defensins, NE, and MPO were also significantly higher in neutrophilic COPD patients and were associated with disease severity [45,131,136]. Additionally, *PAD4* gene expression was significantly higher in neutrophils than in non-neutrophils in COPD patients [45]. This evidence could explain the increased levels of NETs and greater disease severity in the airways of patients with COPD.

### 3.5. Bronchiectasis

Bronchiectasis is a chronic inflammatory disease characterized by mild to moderate airflow obstruction. The pathophysiological hallmark of bronchiectasis is neutrophilic inflammation, which is related to airway infection, lung damage, and impaired mucociliary clearance [137,138]. Neutrophil counts were significantly higher in patients with bronchiectasis than in healthy controls and were positively correlated with the severity of bronchiectasis [139,140]. Neutrophil-derived proteins, including NE, MPO, and MMPs, were also increased in patients with bronchiectasis [54,140]. As a predictor of disease progression, NE showed greater activity in the sputum of patients with bronchiectasis and was associated with exacerbation frequency and decreased FEV1 [141,142,143]. Additionally, increased levels of antimicrobial peptide LL-37 and decreased levels of secretory leucocyte protease inhibitor (SLPI) were associated with decreased FEV1 and exacerbation, indicating that the dysregulation of antimicrobial peptides is associated with disease severity [144]. IL-6 and IL-8 levels were significantly increased in the sputum and BALF of bronchiectasis patients, and neutrophil influx was induced [140,143]. MMP-8 and MMP-9 levels were also significantly higher in patients with bronchiectasis than in healthy controls and were strongly correlated with the progression of bronchiectasis [54,145]. Using sputum proteomics, Keir et al. demonstrated that differentially expressed proteins were associated with components of NETs, including NE, MPO, and neutrophil gelatinase-associated lipocalin (NGAL), and increased amounts of NETs were associated with bronchiectasis severity [20]. These studies demonstrated that NETs contribute to disease severity and lung inflammation.

### 3.6. Coronavirus Disease 2019 (COVID-19)

Coronavirus disease 2019 (COVID-19) is a multisystem inflammatory disease caused by severe acute respiratory syndrome coronavirus 2 (SARS-CoV-2), characterized by distinct patterns of disease progression that suggest diverse host immune responses. An uncontrolled inflammatory response can lead to cytokine storm syndrome and cause severe COVID-19 [146]. Proinflammatory cytokine levels were significantly higher in patients with severe COVID-19 [147,148]. Elevated neutrophils and a neutrophil-to-lymphocyte ratio were observed in patients with COVID-19 compared to healthy controls, predicting disease severity and poor clinical outcomes [148,149,150]. In addition, neutrophils from COVID patients exhibited a larger size and lower granularity than healthy cells [151], and low-density neutrophils were correlated with disease severity [147]. Several studies have reported that NETs were elevated in patients with COVID-19 and were correlated with disease severity [40,152]. In vitro studies have also demonstrated that SARS-CoV-2 induces the release of NETs [152]. NE and MPO levels were higher in the serum of COVID-19 patients [151,153]. eDNA, MPO-DNA, and CitH3 showed significant increases in sera from COVID-19 patients [153]. Serum IL-6, IL-8, and TNF-α levels were also significantly increased in COVID-19 and were associated with severity and poor survival [154].

## 4. Potential Therapeutic Targets of NETs

Therapeutic approaches such as inhibitors of NET formation can help attenuate chronic airway inflammation and reduce the severity of airway diseases, including CRS and asthma (Table 1).

### 4.1. DNase

NETs are degraded by DNase or a DNase-like enzyme targeting eDNA or histones [155,156]. Elevated eDNA has been associated with disease severity in chronic diseases and has become a therapeutic target. DNase I has been approved by the US Food and Drug Administration (FDA) and is commonly used for CF. Recombinant DNase reduces mucus viscosity and improves mucociliary function. DNase treatment improves lung function and reduces exacerbation in airway diseases, such as CF [157,158] and asthma [23,159]. DNase I significantly reduced the expression of MPO and CitH3, and alleviated airway hyper-responsiveness and airway mucus obstruction in a neutrophilic asthma model [92]. Lachowicz-Scroggins et al. demonstrated that NETs damage airway epithelial cells, which can be prevented by disrupting NETs with DNase [23]. DNase also reduced the activity of MPO and NE in CF [113] and COVID-19 patients [160,161]. However, in bronchiectasis studies, it had significant negative effects on patients [162].

### 4.2. Neutrophil Protease Inhibitors

As mentioned for NET formation, NE and MPO are involved in the decondensation of chromatin. Therefore, their inhibition could inhibit NET formation. Endogenous NE inhibitors, such as SERPINA1 (α-1 antitrypsin; AAT) and SLPI, mobilized NE activity. A mouse model with NE deficiency showed significantly reduced mucus hypersecretion and lung damage [163]. AAT deficiency is associated with the prevalence of airway diseases, such as bronchiectasis [164] and CF [165]. SLPI is produced by airway epithelial cells and neutrophils and acts to maintain the protease/antiprotease balance, preventing protease-mediated tissue destruction. SLPI reduces NE activity and IL-8 expression in CF patients [166]. Studies found reduced SLPI levels in patients with exacerbated COPD [167] and in patients with severe asthma [168]. However, levels of the protease inhibitors TIMP-1, SLPI, and elafin were higher in severe COVID-19 patients than in influenza patients [169]. A selective NE inhibitor, GW311616A, prevented ROS production and NET formation and ameliorated lung function in a COPD model both in vitro and in vivo [170]. AZD9668, an oral NE inhibitor, has been tried in COPD [171], CF [172], and bronchiectasis [173]. AZD9668 treatment resulted in decreases in inflammatory markers, such as IL-6 and IL-8, but there was no significant change in NE activity and lung function in bronchiectasis [173] and CF [172]. In clinical trials of COPD patients, there were no significant differences in inflammatory markers, lung function, and patient symptoms [171,174]. An MPO inhibitor, aminobenzoic acid hydrazide (ABAH), delayed NET formation [175]. However, ABAH did not completely inhibit MPO. ABAH reduced airway hyper-responsiveness and oxidative stress in an asthma mouse model with mixed inflammation, but did not inhibit neutrophil and eosinophil infiltration [176]. MPO inhibitors, PF-1355 and AZM198, have been shown to attenuate neutrophil degranulation and NET formation by inhibiting MPO activity in vasculitis [177,178]. Various MPO inhibitors, such as a ferulic acid derivative (INV-315), a thiouracil derivative (OF-06282999), and triazolopyrimidine, are being developed and investigated [179]. As mentioned, although the clinical potential of NE and MPO inhibitors has been investigated, there are limitations such as poor preclinical results, uncertainty regarding the therapeutically effective dose, and potential toxicity. Therefore, more research on various diseases, including airway diseases, is needed.

### 4.3. CXCR2 Antagonists

The chemokine receptor CXCR2 mediates neutrophil migration and induces NET formation through IL-8. AZD5069 is an antagonist of CXCR2, and Pedersen et al. found that it inhibited NET formation in sputum and blood neutrophils from COPD patients [180,181]. AZD5069 has been tried in bronchiectasis [182] and asthma [181,183,184]. In these clinical studies, neutrophil counts were reduced, but AZD5069 treatment did not improve clinical outcomes. In a clinical study of CF with the selective CXCR2 antagonist SB-656933, neutrophils, NE, and free DNA levels were reduced, while IL-8 and fibrinogen levels were increased. However, there were no changes in lung function and symptoms, and side-effects occurred in some patients [185]. Danirixin demonstrated antagonism of CXCR2 activity and blockade of mediated neutrophil activation both in vitro and in vivo [186]. Clinical trials for COPD have shown no clinical benefits for patients [187,188]. Reparixin, an IL-8 receptor (CXCR1/2) blocker, inhibits CXCL8-induced neutrophil activation. Reparixin treatment significantly reduced neutrophil recruitment in a mouse model of LPS-induced pulmonary inflammation [189]. Neutralizing IL-8 using reparixin and an anti–IL-8 antibody reduced NET formation in the plasma of COVID-19 patients [67].

### 4.4. ROS Scavengers

ROS production is an important inducer of NET formation. Diphenyleneiodonium chloride (DPI) inhibits gluconeogenesis and oxidative stress by inhibiting NADPH oxidase. DPI also inhibits eDNA release and NET formation [190,191,192]. Nevertheless, N-acetyl-L-cysteine (NAC) and DPI decrease inflammatory cytokine and ROS production, thereby inhibiting EET formation and improving lung function in asthma [193,194]. NAC, an antioxidant, decreases mucous viscosity and indirectly inhibits NET formation. NAC treatment reduced PMA-induced NETs, but H_2_O_2_- and *Staphylococcus aureus*-induced NETs did not change [195]. NAC reduced the risk of acute exacerbations in patients with COPD [196,197]. In patients with bronchiectasis, NAC treatment was also associated with a reduced incidence of exacerbations and sputum production, with no severe adverse effects [198].

### 4.5. Other Inhibitors

Inhibition of PAD4 prevents NET formation by reducing histone citrullination [199,200,201]. Cl-amidine, a PAD4 inhibitor, reduced the neutrophil count, decreased levels of citrullinated histone H3 (CitH3) and inflammatory cytokines, and inhibited NET formation and tissue damage [200,201,202]. This agent also inhibited EET formation [203,204]. Streptonigrin is also a selective inhibitor of PAD4 that led to decreased expression of CitH3 and proinflammatory cytokines [205]. A novel small-molecule PAD4 inhibitor (PAD4i) developed by AstraZeneca reduced eDNA, CitH3, and NET formation in *Haemophilus influenzae*-infected or ionomycin/calcium chloride-stimulated neutrophils [206].

Hydroxychloroquine (HCQ), an FDA-approved antimalarial drug, is a less potent derivative of chloroquine that inhibits activated neutrophils and NET formation by attenuating Toll-like receptor-9 (TLR-9) and ROS production [207]. It has also been reported that HCQ reduced the levels of proinflammatory cytokines [208], but did not affect the expression of NE, PAD4, and MPO [209]. Several studies have proven that HCQ attenuated NET formation in patients with systemic lupus erythematosus [210] and COVID-19 [211].

Metformin is one of the most widely prescribed antidiabetic drugs. Metformin activates AMP-activated protein kinase (AMPK) and inhibits the mTOR pathway. Metformin reduced neutrophil counts and NET formation in several diseases [212,213,214]. Metformin reduced the expression of NET components, such as NE, histones, and eDNA, as well as prevented PMA- and ionomycin-induced NET formation through inhibition of NADPH oxidase [214]. Several studies have shown that metformin reduced markers of inflammation and may be associated with attenuation of mortality and poor outcomes from COVID-19 [215,216]. However, Oh et al. reported that metformin reduced the risk of COVID-19 in patients with type 2 diabetes mellitus, but did not affect mortality [217].

In addition, there are many studies on various phytochemicals that inhibit NET formation. *Epilobium pyrricholophum* extract [218] and *Artemisia gmelinii* [219] extract significantly attenuated inflammatory cell recruitment, including neutrophil, and reduced IL6 and IL8 expression in a COPD mouse model. Moreover, *Artemisia gmelinii* extract also protected lung tissues from injury [219]. Monomeric epigallocatechin-3-gallate, a component of green tea, inhibited NE activity and reduced NET formation and tissue damage in vivo [220]. Gingerols [221] and zingerone [222], compounds derived from ginger, attenuates NET formation and ROS production. However, more research is needed to prove the therapeutic effects of phytochemicals in airway diseases.
ijms-24-05034-t001_Table 1Table 1An overview of potential targets of neutrophil extracellular traps (NETs) in airway diseases.Target MoleculeInhibitor(s)Clinical TrialDisease(s)Reference(s)DNADNase IFDA approvedAsthma[23,92,159]CF[113,157,158]Bronchiectasis[162]COVID-19[160,161]Neutrophil elastase (NE)AZD9668Phase IIaCF[172]Phase IIbCOPD[171,174]Phase IIBronchiectasis[173]GW311616ANACOPD[170]Myeloperoxidase (MPO)ABAHNAAsthma[176]CXCR2AZD5069Phase IIbAsthma[183,184]Phase IIa [223] COPD[180]Phase IIaBronchiectasis[182]DanirixinPhase IIbCOPD[187,188]CXCR1/2ReparixinPhase II/III(COVID-19 pneumonia) [224] COVID-19[67]Reactive oxygen species (ROS)N-acetyl-L-cysteine (NAC)FDA approvedCOPD[196,197]
Bronchiectasis[198]Multiple pathwaysHydroxychloroquine (HCQ)FDA approvedCOVID-19[211]MetforminFDA approvedCOVID-19[215,216,217]

## 5. Conclusions

Neutrophils are an essential component of the innate immune system and play an important role in airway diseases including CRS. In addition, neutrophils and NETs are involved in the progression and severity of several airway diseases. This review focused on understanding the mechanisms underlying NET formation and suggesting NET inhibitors. DNase, NE inhibitors, and ROS scavengers have been proposed as therapeutic approaches to inhibit NET formation, but limitations such as therapeutic efficacy or toxicity in preclinical or clinical stages remain to be resolved. Although many studies have been conducted on the pathophysiological relevance of NET formation, further research is still needed to investigate novel treatment strategies for airway diseases.

## Figures and Tables

**Figure 1 ijms-24-05034-f001:**
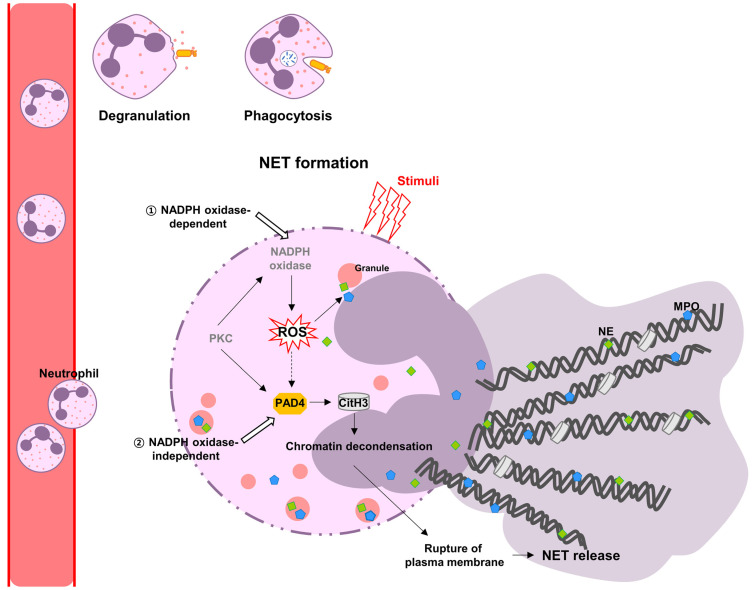
NADPH oxidase-dependent or -independent NET formation. Neutrophils eliminate invading pathogens through phagocytosis, degranulation, and NET formation. ① NADPH oxidase-mediated ROS stimulates MPO and NE to promote translocation from neutrophil granules into the nucleus and trigger the release of NETs. MPO binds chromatin with NE and contributes to the decondensation of chromatin, and then the nuclear membrane is disrupted. ② Activated PAD4 citrullinates histones, causing chromatin decondensation. NETs are released into the extracellular space to capture invading pathogens.

**Figure 2 ijms-24-05034-f002:**
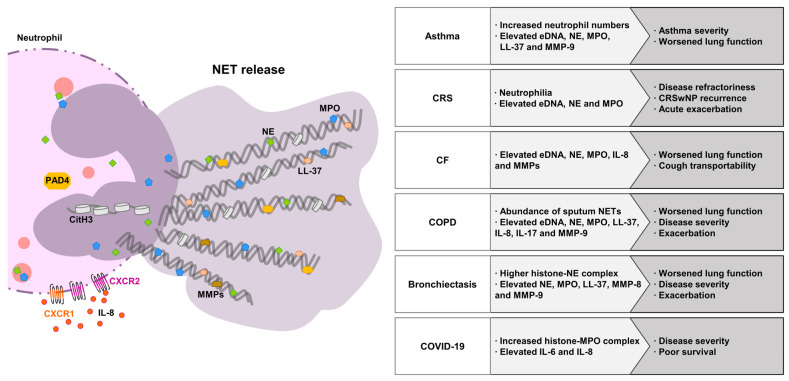
Clinical implications for neutrophil extracellular trap (NET) components in airway diseases. Elevated NET components have been implicated in the pathogenesis of airway diseases. NET: neutrophil extracellular trap; eDNA: extracellular DNA; NE: neutrophil elastase; MPO: myeloperoxidase; PAD4: protein arginine deiminase 4; CitH3: citrullination of histone H3; IL: interleukin; MMP: matrix metalloproteinase; CXCR: C–X–C motif chemokine receptor; CRS: chronic rhinosinusitis; CF: cystic fibrosis; COPD: chronic obstructive pulmonary disease; COVID-19: coronavirus disease 2019.

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
