# Peer review of "Neutrophil Extracellular Traps in Airway Diseases: Pathological Roles and Therapeutic Implications"

_ijms, 2023, doi:10.3390/ijms24055034_

Round 1

Reviewer 1 Report

In the review, the authors drew attention to a very interesting and urgent topic. This is the role of extracellular neutrophil traps in inflammatory lung diseases. NETs, like other sources of reactive oxygen species, are a double-edged sword, so one should very clearly distinguish between harmful and beneficial effects of them. The development of therapeutic strategies for the control of NETs is very promising and of practical value. The review is well structured, contains summarizing tables and a large number of up-to-date references. I have two small comments:

1. Please check for transcripts before abbreviations. For example, ROS and PKC.

2. The paragraph of line 46-88 seems too long to me. It contains heterogeneous information about different neutrophil enzymes. It seems to me that for ease of perception it should be divided into several paragraphs.

Author Response

Reviewer #1 (Comments and Suggestions for Authors):

In the review, the authors drew attention to a very interesting and urgent topic. This is the role of extracellular neutrophil traps in inflammatory lung diseases. NETs, like other sources of reactive oxygen species, are a double-edged sword, so one should very clearly distinguish between harmful and beneficial effects of them. The development of therapeutic strategies for the control of NETs is very promising and of practical value. The review is well structured, contains summarizing tables and a large number of up-to-date references. I have two small comments:

  1. Please check for transcripts before abbreviations. For example, ROS and PKC.

Response: Thank you for pointing this out. We have added for the first abbreviations (Please see page 2, line 59 (NADPH), line 60 (ROS) and line 91 (PKC); highlighted with yellow color).

  1. The paragraph of line 46-88 seems too long to me. It contains heterogeneous information about different neutrophil enzymes. It seems to me that for ease of perception it should be divided into several paragraphs.

Response: As reviewer’s suggestion, we have divided it into two more paragraphs. (Please see page 2).

Reviewer 2 Report

The presented review paper focuses on the biology of NETs and possible pharmacological interventions for their breakdown.

The discussed topic is an interesting and still gaining more and more attention in the field of immunology and hemostasis, I like the structure, the flow of the draft, however I have some comments.

In general, I have two very major comments, that must be addressed and fixed before processing further.

Please follow my two points.

Firstly, the aim of the review paper is to discuss in a comprehensive manner the recent advances in the discussed area. Here, I see lots of references dated 2005-2015. Recently, basic databases like W-O-S or PubMed shows over 300 records for nets+neutrophils+and similar searches. Only a few are included in the review. The novelty and very narrow literature search is the major problem that must be fixed.

Moving further (even for narrative reviews), there is a very limited amount of information given regarding the following PRISMA guidelines and description of the searching strategy, which is so crucial for review papers. The Author should include at least: Data sources and searches, Study eligibility criteria, Study selection process, Data extraction, and study quality assessment (assessing the risk of bias (ROB) for each included study), Data synthesis. MeSH terms (in addition/replacement of keywords) are necessary to be included. For each step, it is necessary to explain to the reader with pictures or tables. It is necessary to explain what was drawn at each step to lead to the result. Moreover, a figure showing the PRISMA-based workflow must be drawn accordingly to the Prisma schema. After that, a discussion is valuable even for narrative papers. A description of the Data Mining strategy should also be included.

Recently, there were published two dogma-shift papers in the area of NETs - please discuss these two extremely relevant papers:

https://doi.org/10.1182/blood.2021014552

https://doi.org/10.1038/s41420-022-01166-3

I would also suggest creating the figure which will bring some new mechanistic approaches to the process of NETosis - the presence of good figure sometimes says more than 1000 words.

There are some minor grammatical errors / minor typos - please go over them carefully.

Please also re-format the tables at the end - there are taking too much space.

Author Response

Response: The authors would like to thank the Reviewer for their comments.

Reviewer 3 Report

The review article by Jo and Kim provides an overview of NETs in pulmonary diseases. Overall review is comprehensive and well written. There are some issues that the authors should consider before its possible publication in IJMS.

1.     Authors should include few more sentences related to neutrophils functions in the introduction part (line 21-24).

2.     Several key references are missing for example Uddin et al. 2019, Porto et al. 2016, Keir and Chalmers 2022, Trivedi et al. 2021, etc.,

3.     Expand ROS in first appearance (line 57).

4.     Mention the phytochemicals/small molecules against NETs during airway inflammation/COPD.

5.     Discuss the role of ROS and NADPH induced NET formation in more detail.

6.     To me the major drawback of this review article is not including any figures depicting the overall flow. I strongly suggest to include two figures one is for the general NETs formation and other one for mechanistic involvement of NETs in airway diseases.

Author Response

Reviewer #3 (Comments and Suggestions for Authors):

The review article by Jo and Kim provides an overview of NETs in pulmonary diseases. Overall review is comprehensive and well written. There are some issues that the authors should consider before its possible publication in IJMS.

  1. Authors should include few more sentences related to neutrophils functions in the introduction part (line 21-24).

Response:  Thank you for this suggestion. We have revised that point in Introduction section. (Page 1, line 22 to 25; newly added sentences are highlighted with yellow color)

  1. Several key references are missing for example Uddin et al. 2019, Porto et al. 2016, Keir and Chalmers 2022, Trivedi et al. 2021, etc.,

Response: As reviewer’s suggestion, we cited this review article in 3. NETs in airway disease section  (Ref 74: Keir and Chalmers. Eur Respir Rev. 2022; Ref 75: Porto and Stein. Front Immunol. 2016; Ref 76: Trivedi et al. Biomedicines. 2021) and 4-3. CXCR2 antagonists section (Ref 181: Uddin et al. Front Immunol. 2019).

  1. Expand ROS in first appearance (line 57).

Response: Thank you for pointing this out. We have added for the first abbreviations (Please see the page 2, line 60; highlighted with yellow color).

  1. Mention the phytochemicals/small molecules against NETs during airway inflammation/COPD.

Response: As reviewer’s suggestion, we have revised that point in 4-5. Other inhibitors. (Page 8, line 356 to 358 and line 374 to 382; newly added sentences are highlighted with yellow color). Small molecules (e.g., CXCR2 antagonists; AZD5069, danirixin) have already been mentioned, so the study on PAD4i was added.

  1. Discuss the role of ROS and NADPH induced NET formation in more detail.

Response: As reviewer’s suggestion, we have updated our manuscript for this in 2. Neutrophil extracellular traps (NETs) section (Please see page 2, line 59 to 65;  highlighted with yellow color).

  1. To me the major drawback of this review article is not including any figures depicting the overall flow. I strongly suggest to include two figures one is for the general NETs formation and other one for mechanistic involvement of NETs in airway diseases.

Response: Thank you for this suggestion. We agree with this comment. Thus, we replaced Table 1 with Figure 1 to show the potential targets of neutrophil extracellular traps (NETs) in airway diseases.

Reviewer 4 Report

It is an important to understand the role of NETs in airway diseases. With enough references, the authors had a great job for organizing and analyzing these publications to understand the role of neutrophil extracellular trap in airway diseases. I prefer to endorse this review in present form.   

Author Response

(The authors gave the same response as above.)

Round 2

Reviewer 2 Report

The Authors addressed all my comments in a satisfactory way. 

The only thing is my suggestion to put the PRISMA graph in the supplementary file since it gives overall insights into the search strategy.

Author Response

Reviewer #2 (Comments and Suggestions for Authors):

The Authors addressed all my comments in a satisfactory way.

The only thing is my suggestion to put the PRISMA graph in the supplementary file since it gives overall insights into the search strategy.

Response: Thank you for pointing this out. As reviewer’s suggestion, we have updated this point in the supplementary.

Reviewer 3 Report

Authors have satisfactorily answered the queries.

But I don't see the figure in the revised version as against their claim in the response letter (Table 1 changed to figure1).

Author Response

Reviewer #3 (Comments and Suggestions for Authors):

Authors have satisfactorily answered the queries.

But I don't see the figure in the revised version as against their claim in the response letter (Table 1 changed to figure1).

Response: As reviewer’s suggestion, We have revised the figure 1 and figure 2 and described additional explanation in the figure legends. (Please see the page 10, line 404 to 420; newly added sentences are highlighted with yellow color).